

# On optimizing the sensor spacing for pressure measurements on wind turbine airfoils

Erik K. Fritz[1], Christopher L. Kelley[2], and Kenneth A. Brown[2]

[1]Wind Energy, TNO Energy Transition, Petten, Netherlands
[2]Sandia National Laboratories, Albuquerque, United States of America

**Correspondence:** Erik Fritz (e.fritz@tno.nl)

**Abstract.** This research article presents a robust approach to optimizing the layout of pressure sensors around an airfoil. A genetic algorithm and a sequential quadratic programming algorithm are employed to derive a sensor layout best suited to represent the expected pressure distribution and, thus, the lift force.

The fact that both optimization routines converge to almost identical sensor layouts suggests that an optimum exists and is reached. By comparing against a cosine-spaced sensor layout, it is demonstrated that the underlying pressure distribution can be captured more accurately with the presented layout optimization approach. Conversely, a 39-55 % reduction in the number of sensors compared to cosine spacing is achievable without loss in lift prediction accuracy. Given these benefits, an optimized sensor layout improves the data quality, reduces unnecessary equipment and saves cost in experimental setups.

While the optimization routine is demonstrated based on the generic example of the IEA 15 MW reference wind turbine, it is suitable for a wide range of applications requiring pressure measurements around airfoils.

## 1 Introduction

Pressure measurements are an essential technique in analysing the flow over aerodynamic bodies. By having knowledge of the pressure field distributed over an airfoil surface, flow characteristics can be determined, and aerodynamic forces can be derived. Pressure measurements are, therefore, well established throughout different research communities, such as aircraft engineering (Barlow et al., 1999) and wind turbine engineering (Schreck, 2022).

Most commonly, they are used to derive airfoil polars, thus, the non-dimensionalized aerodynamic forces and moments as a function of inflow angle of attack (Timmer and Rooij, 2003; Post et al., 2008; Coder and Maughmer, 2014; Pires et al., 2016; Bartl et al., 2019; Holst et al., 2019b; Brunner et al., 2021). Of particular interest to the wind energy sector, where airfoils rotate and experience different inflow conditions throughout one rotation, is the determination of unsteady airfoil polars (Lee and Gerontakos, 2004; Holst et al., 2018, 2019a; Mayer et al., 2020; De Tavernier et al., 2021).

Modern wind turbines make use of a variety of blade add-ons to improve local blade aerodynamics. Surface pressure measurements can be used to study the changes in local airfoil aerodynamics imposed by add-ons such as Gurney flaps (Cole et al., 2013; Balduzzi et al., 2021), vortex generators (Baldacchino et al., 2018) or trailing edge flaps (Bak et al., 2010; Madsen et al., 2022). In the latter case, pressure measurements have also been used as input for actuation control of trailing edge flaps



(Gaunaa and Andersen, 2009; Velte et al., 2012; Bartholomay et al., 2021). Other application areas include investigations into boundary layer transition behaviour (Groenewoud et al., 1983; Schaffarczyk et al., 2016) or the use of surface pressure spectra for noise modelling (Bertagnolio et al., 2017).

In larger experimental setups on rotating blades, blade aerodynamics can be characterized by measuring pressure distributions at multiple radial locations (Butterfield et al., 1992; Brand et al., 1996; Bruining, 1997; Simms et al., 1999; Hand et al.,

2001; Schepers et al., 2002; Maeda and Kawabuchi, 2005; Schepers and Snel, 2007; Bak et al., 2010, 2011; Medina et al., 2012; Boorsma and Schepers, 2015).

Finally, a critical application of such measurements lies in creating reference datasets that can be used for numerical model validation (Singh et al., 2012; Sarlak et al., 2014; Heißelmann et al., 2016; Schepers and Snel, 2007; Boorsma and Schepers, 2015).

Irrespective of the application, the amount of sensors and their placement on the airfoil's surface impacts the accuracy with which the aerodynamic properties of the airfoil can be characterized. A logical consensus is that the pressure sensors should be more densely placed towards the airfoil's leading edge to capture the higher gradients in the pressure distribution commonly present in this region. While some authors mention this explicitly (Butterfield et al., 1992; Simms et al., 1999; Hand et al., 2001; Maeda and Kawabuchi, 2005; Holst et al., 2018), the same can be derived for most other studies mentioned above based

on the published graphs/schematics. Very few authors go beyond this level of detail regarding the thought process that went into the sensor layout. Brunner et al. (2021) gave a mathematical formulation to derive the sensor spacing, which ensures higher resolution at the leading edge. Bak et al. (2010) state that "the distribution of the pressure taps was decided from the theoretical target pressure distributions to reflect the expected pressure gradients". While indicating a more strategic approach to determining the layout, unfortunately, no further details are given.

The lack of detail regarding the selected pressure sensor layout shows that, in most cases, this issue is tackled by simply using a very high number of pressure taps, resulting in an apparently high enough resolution of the pressure distribution. There exist, however, many situations where this is not possible. Limitations on the number of available sensors could be imposed by geometrical considerations, such as small-scale experimental geometries or the use of airfoils with internal structures, structural concerns where too many sensors endanger safe operation, or simply the sensor price. The latter is becoming especially

relevant as new sensor technologies such as fibre optical pressure sensors pose an alternative to the historically most common arrangement of pressure taps leading to transducers. Furthermore, it can be desirable to limit the number of sensors to minimize flow disturbances that could trip the boundary layer or alter measurements further downstream. For such situations, wherein the number of available/allowable sensors is limited, there is a need for a robust approach to finding an optimal sensor spacing which represents the airfoil's pressure distribution and, thus, aerodynamic characteristics as accurately as possible.

In the present work, two optimization routines (genetic algorithm and sequential quadratic programming) are used to derive the optimal pressure sensor layout for various airfoils. While applied to the case of rotating wind turbine airfoils, the approach is suited just as well for aerospace applications or wind tunnel experiments. In this study, the sensor layout is optimized for a range of angles of attack, where each angle is weighted based on its probability of occurrence. Results of the optimized pressure sensor layouts are compared against a simple cosine sensor spacing, which is closer to the sensor layouts used in current



experiments. Based on the accuracy of lift prediction and the ability to closely represent the expected pressure distribution, the potential to reduce the number of sensors is studied.

This article is built up as follows: Section 2.1 introduces the airfoils selected for this study and their expected operating conditions. The airfoil polars used as input for the optimization routine are presented in section 2.2. Section 2.3 details the equations to determine the error in load estimation. Section 2.4 introduces the sensor layout optimization routines as well as the approach of cosine spacing serving as reference. Section 3.1 presents the accuracy in load estimation that can be achieved when applying cosine sensor spacing. Building on this, the improvement in accuracy when using an optimized sensor layout is demonstrated in section 3.2. Section 3.3 discusses the potential of reducing the number of sensors without losing accuracy by layout optimization. Finally, the findings of this investigation are summarized in section 4 and concluding remarks are given.

## 2 Methodology

### 2.1 Selected airfoils and their operating conditions

For the present study, the IEA 15 MW reference wind turbine (RWT) is chosen. All relevant information is taken from the report by Gaertner et al. (2020) and the complimentary GitHub repository (Barter et al., 2023). The IEA 15 MW RWT's blade is defined using the FFA airfoil family. A schematic of the blade geometry, along with the starting positions of the respective airfoils, is shown in figure 1. This study focuses on the four most outboard, non-blended airfoils: FFA-W3-360, FFA-W3-301, FFA-W3-241 and FFA-W3-211.

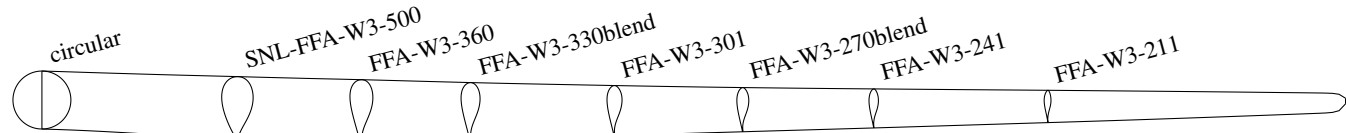

**Figure 1.** IEA 15 MW RWT blade and the starting locations of the airfoils used in the blade definition

The information included in the IEA 15 MW documentation is used to estimate the operating conditions of the respective airfoils in a simplified approach. The turbine is categorized as turbine class IB as defined in IEC standard 61400-1 (International Electrotechnical Commission, 2005). According to this standard, the normal wind conditions experienced by a wind turbine are given by a Rayleigh distribution with cumulative distribution function

$$CDF(U_\infty) = 1 - \exp\left(-\pi\left(\frac{U_\infty}{2U_{ave}}\right)^2\right) \tag{1}$$

and probability density function

$$PDF(U_\infty) = \frac{\pi U_\infty}{2U_{ave}^2}\exp\left(-\pi\left(\frac{U_\infty}{2U_{ave}}\right)^2\right) \tag{2}$$





where $U_\infty$ is the wind speed at hub height and $U_{ave}$ is defined as $U_{ave} = 0.2 U_{ref}$. The reference wind speed $U_{ref}$ is defined per turbine class, in the case of IEC class IB $U_{ref} = 50$ m/s. Figure 2 shows the Rayleigh probability density function between

the cut-in and cut-out wind speed of the IEA 15 MW RWT.

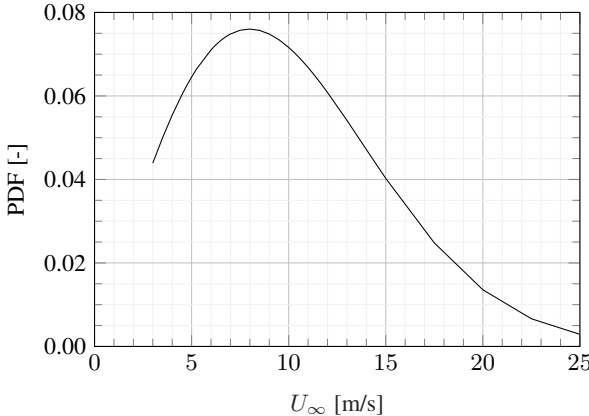

**Figure 2.** Rayleigh wind distribution according to IEC 61400-1 for turbine class IB

Now, the documented rotor performance data (Barter et al., 2023) is used to estimate the operating regime of the blade cross sections under investigation. Applying 1D momentum theory with Glauert correction for heavily loaded rotors (see e.g. Burton et al., 2011), the rotor averaged induction factor $a$ is calculated as a function of the thrust coefficient $C_T$, which is given in the turbine documentation for the operating range of wind speeds.

$$
a = \begin{cases} \frac{1}{2} - \frac{\sqrt{1 - C_T}}{2}, & \text{for } C_T < C_{T_2} \\ 1 + \frac{C_T - C_{T_1}}{4\sqrt{C_{T_1}} - 4}, & \text{for } C_T \geq C_{T_2} \end{cases}
\tag{3}
$$

where $C_{T_1} = 1.816$ and $C_{T_2} = 2\sqrt{C_{T_1}} - C_{T_1}$. By applying the Prandtl root and tip corrections

$$
F_{tip} = \frac{2}{\pi} \cos^{-1} \left( e^{-\frac{N_b}{2}\left(\frac{R}{r} - 1\right)\sqrt{1 + \left(\frac{\lambda_r}{1-a}\right)^2}} \right)
\tag{4}
$$

$$
F_{root} = \frac{2}{\pi} \cos^{-1} \left( e^{\frac{N_b}{2}\left(\frac{r_{root}}{r} - 1\right)\sqrt{1 + \left(\frac{\lambda_r}{1-a}\right)^2}} \right)
\tag{5}
$$

where $r_{root}$ and $R$ are the root and tip radius and $\lambda_r$ is the local tip speed ratio, the rotor averaged induction factor can be

converted to a local blade induction factor $a_B = \frac{a}{F_{tip} F_{root}}$. Now, the local inflow angle can be calculated as

$$
\phi = \tan^{-1} \left( \frac{U_\infty (1 - a_B)}{\omega r (1 + a'_B)} \right)
\tag{6}
$$

where $\omega$ is the angular velocity. Given that the investigated airfoils are located in spanwise regions where tangential induction is expected to have little impact, it is assumed to be $a'_B = 0$. Consequently, the angle of attack is calculated as

$$
\alpha = \phi - \beta_{twist} - \beta_{pitch}
\tag{7}
$$




where $\beta_{twist}$ is the local blade twist angle and $\beta_{pitch}$ is the global blade pitch angle. Equation 7 neglects elastic twist deformations that should be considered if reliable data or simulation results are available. The angles of attack estimated through this simplified approach are shown for the investigated airfoils as a function of the wind speed in figure 3. In realistic conditions, environmental/operational conditions, such as turbulence or shear, would lead to a range of angles of attack present for each wind speed.

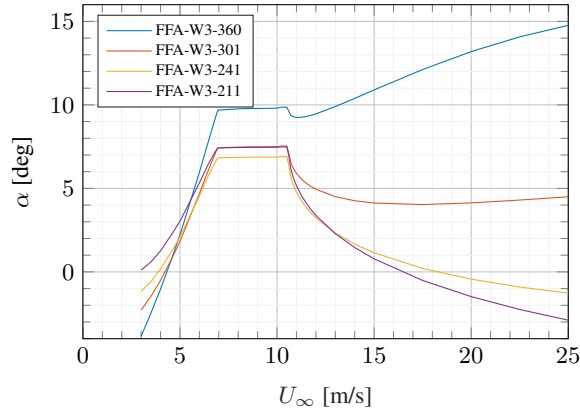

**Figure 3.** Angle of attack as function of wind speed

## 2.2 Generating airfoil polars using XFOIL

Airfoil polars and corresponding pressure distributions are prerequisites for the sensor layout optimization approaches presented in sections 2.4.2 and 2.4.3. In this study, the 2D viscous/inviscid code XFOIL, developed by Drela (1989), is used to generate these polars. When simulating viscous airfoil polars, this code requires the chord Reynolds number $\mathrm{Re}_c$ as input. It is defined as

$$\mathrm{Re}_c = \frac{\rho V_{eff} c}{\mu} \tag{8}$$

where $\rho$ and $\mu$ are the density and dynamic viscosity of air, respectively. The local effective velocity can be calculated as

$$V_{eff} = \sqrt{\left(U_\infty \left(1 - a_B\right)\right)^2 + \left(\omega r \left(1 + a'_B\right)\right)^2} \tag{9}$$

At the IEA 15 MW RWT's rated wind speed $U_\infty = 10.59 \, \mathrm{m/s}$, the thrust coefficient is $C_T = 0.769$ and the rotor speed is $\omega = 7.56 \, \mathrm{rpm}$, resulting in a tip speed ratio of $\lambda = 8.97$, see Barter et al. (2023). Using the approach detailed in section 2.1, 115 the rotor averaged axial induction factor and, consequently, the local blade axial induction are determined. Again, tangential induction is assumed to be negligible. The approximated chord Reynolds numbers are listed alongside geometric information of the airfoils in table 1. Here, the properties of air are assumed as $\rho = 1.204 \, \mathrm{kg/m^3}$ and $\mu = 1.825 \, e^{-5} \, \mathrm{kg/(m\,s)}$, corresponding to $20 \, {}^\circ\mathrm{C}$ and standard atmospheric pressure.





| Airfoil | $r$ [m] | $r/R$ [-] | $c$ [m] | $t/c$ [-] | $\mathrm{Re}_{c,approx}$ [-] |
|---|---|---|---|---|---|
| FFA-W3-360 | 31.68 | 0.26 | 5.70 | 0.360 | 9.86 e6 |
| FFA-W3-301 | 54.38 | 0.45 | 4.48 | 0.301 | 12.93 e6 |
| FFA-W3-241 | 77.67 | 0.65 | 3.50 | 0.241 | 14.31 e6 |
| FFA-W3-211 | 93.29 | 0.78 | 2.90 | 0.211 | 14.20 e6 |

**Table 1.** FFA airfoils as used in the definition of the IEA 15 MW RWT and their approximated chord Reynolds number

Based on the approximated chord Reynolds numbers, the airfoil polars are simulated. The results generated with XFOIL are depicted in figure 4. Given the expected angles of attack as shown in figure 3, the polars are determined between $\alpha = -5°$ and $\alpha = 15°$ with a step size of $\Delta\alpha = 0.25°$. To mimic turbulent inflow conditions likely to occur for a wind turbine in the field, boundary layer transition is enforced at $x/c = 0.05$ on the suction side and at $x/c = 0.1$ on the pressure side. The XFOIL simulations were run using 160 panels to discretize the airfoils, with the exception of the FFA-W3-211 airfoil, which was simulated using 195 panels to avoid convergence issues.

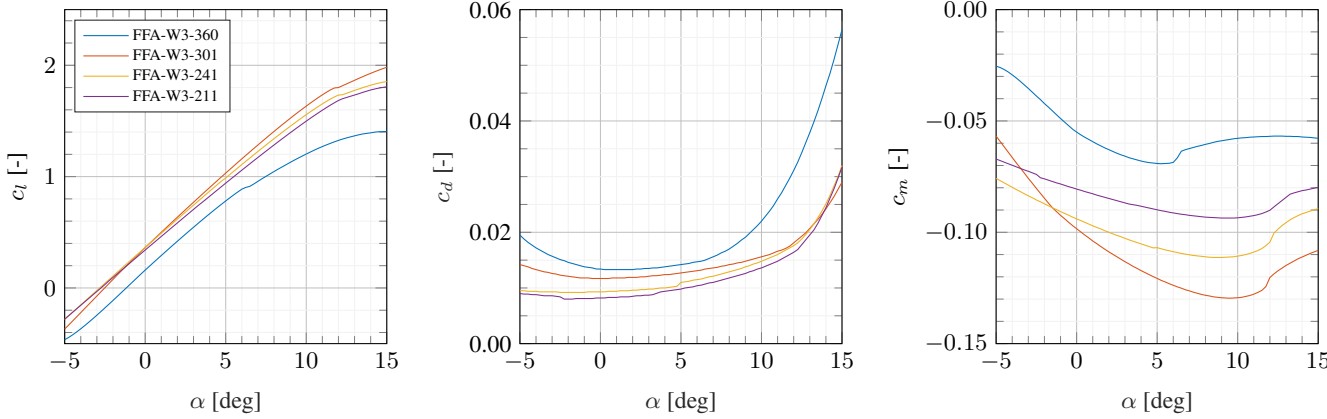

**Figure 4.** Airfoil polars as simulated by XFOIL

It should be noted that XFOIL is one way of generating the polars and pressure distributions later used as inputs for the optimization routine. This code was chosen for its widespread use and open access. Alternatively, the required data could be obtained using other approaches, e.g. RFOIL, which is an adaptation of XFOIL developed for rotating airfoils (Van Rooij, 1996; Ramanujam et al., 2016), or higher fidelity tools such as computational fluid dynamics (CFD).

### 2.3 Estimating lift based on a discrete number of pressure sensors

The polar curves presented in the previous section correspond to the forces distributed over the airfoil surface. Based on the surface pressure coefficient distribution $c_p$, the chord normal force coefficient $c_n$ and chord tangential force coefficient $c_t$ are





calculated as

$$
\begin{bmatrix} c_t \\ c_n \end{bmatrix} = \int_S c_p(s) \cdot \boldsymbol{n}(s) \, \mathrm{d}s
\tag{10}
$$

where $\boldsymbol{n}$ is the surface normal vector and $s$ is the surface coordinate. It should be realized that these forces do not account for
forces due to skin friction. Skin friction forces typically represent a negligible contribution to the lift and pitching moment.

The lift coefficient can be determined by decomposing the normal and tangential force coefficients

$$
c_l = c_n \cos(\alpha) - c_t \sin(\alpha)
\tag{11}
$$

In an experimental setup, information regarding the surface pressure is only available at the discrete points on the airfoil
surface where pressure sensors are placed. These discrete points can then be interpolated to derive a pressure distribution
spanning the entire airfoil surface. How accurate this interpolation and, thus, the integrated airfoil loads are depends on the
number and placement of sensors used. Additionally, a chosen sensor layout might not be equally suitable for all angles of
attack. Therefore, one should consider whether priority is given to optimally resolving the pressure distribution for

1. a single angle of attack,

2. a range of angles of attack given equal priority, or

3. a range of angles of attack weighted based on their likelihood to occur during operation/testing.

In the first case, the error between the lift coefficient determined based on the pressure distribution interpolated between
sensor locations $c_{l,int}$ and the expected true value of the airfoil coefficient $c_{l,exp}$ is simply their difference

$$
E\left(c_l\right) = \left(c_{l,int}(\alpha) - c_{l,exp}(\alpha)\right)
\tag{12}
$$

When giving equal priority to several angles of attack $N_\alpha$, the error between interpolated and expected lift coefficient can
be expressed as the mean error

$$
\bar{E}\left(c_l\right) = \frac{1}{N_\alpha} \sum_{\alpha=\alpha_{min}}^{\alpha_{max}} \left(c_{l,int}(\alpha) - c_{l,exp}(\alpha)\right)
\tag{13}
$$

In the present study, the third variant is used. Combining the wind speed distribution shown in figure 2 with the expected
angle of attack shown in figure 3, the probability of the occurrence of an angle of attack can be calculated. For this purpose,
the expected angles of attack are binned using the angle of attack discretisation used in the XFOIL simulations. The resulting
probabilities are given in figure 5, where the spikes are due to the binning of the angles of attack.

Now, the probability-weighted error in the prediction of the lift coefficient based on the measurements of a discrete number
of pressure sensors can be calculated as

$$
E_{prob}\left(c_l\right) = \frac{1}{C_{PDF}} \sum_{\alpha=\alpha_{min}}^{\alpha_{max}} P(\alpha)\left(c_{l,int}(\alpha) - c_{l,exp}(\alpha)\right)
\tag{14}
$$



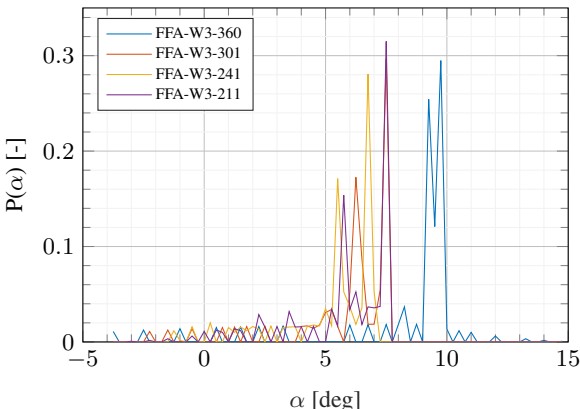

**Figure 5.** Probability of occurrence of an angle of attack for the investigated airfoils

where $P(\alpha)$ is the probability of an angle of attack to occur. Because the integral of the probability density function shown
in figure 2 is not equal to unity between the cut-in and cut-out speed, a scaling factor $C_{PDF} = \int_{U_{cut-in}}^{U_{cut-out}} PDF(U_\infty)\mathrm{d}U_\infty$ is
applied to the weights. This ensures that the scaled sum of probabilities equals unity and the weighted error is representative
of an actual deviation in lift coefficient.

### 2.4 Approaches to define the pressure sensor layout

#### 2.4.1 Cosine spacing

There is consensus in the literature that the pressure sensor layout should be most dense where high gradients in the pressure
distribution need to be resolved. Most commonly, this entails the highest sensor density at the airfoil's leading edge, where
pressure gradients are the largest of any location on the airfoil, and trailing edge, where the onset of trailing-edge flow separation
similarly can produce relatively large local gradients. An easy way to create such a sensor layout is by applying a cosine
distribution as shown in figure 6 for $N_s = 15$ sensors on the FFA-W3-241 airfoil.

#### 2.4.2 Genetic algorithm layout optimization (GA)

Genetic algorithms imitate biological evolutionary behaviour, and their functionality is only briefly summarized here in a
simplified manner: In the initial iteration, a population of random design variable sets is generated. Based on a rating of their
fitness, thus, their ability to minimize the objective function, "parent variable sets" are chosen from which "children variable
sets" are generated that form the population of the next iteration. This evolutionary process is repeated until a convergence
criterion is met.

In this study, the design variables are the sensor positions of $N_s$ pressure sensors $p_i$ with $i \in [1, 2, ..., N_s]$. Each design
variable is bounded by $0 \le p \le 2$ where $p$ is the coordinate along the chord line moving from the trailing edge of the suction
side ($p = 0$) to the leading edge ($p = 1$) and back via the chord line to the trailing edge of the pressure side ($p = 2$). Each





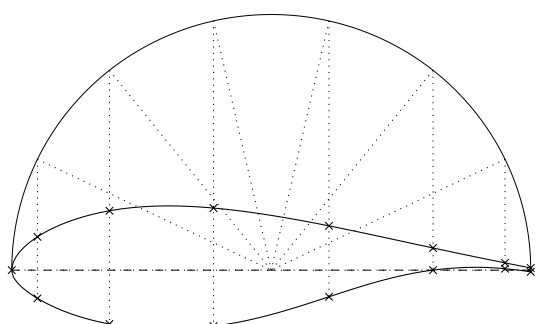

**Figure 6.** Sensor layout using a cosine spacing approach on the FFA-W3-241 airfoil, $N_s = 15$

population generation consists of 5000 sets of $N_s$ sensor positions, and the convergence criterion is met when 15 consecutive
generations do not result in an improvement of fitness. The objective function is chosen as

$$\min E_{prob}(c_p) = \frac{1}{C_{PDF}} \sum_{\alpha=\alpha_{min}}^{\alpha_{max}} P(\alpha) \int_S |c_{p,int}(\alpha,s) - c_{p,exp}(\alpha,s)| \, \mathrm{d}s \tag{15}$$

which targets an optimal match between the expected and interpolated pressure distribution. The absolute values of their local
difference are used to avoid the cancellation of errors, e.g. an equivalent shaving of the negative suction peak and the positive
stagnation peak. For the same reason of error cancellation, it is not advisable to directly optimize for a minimal error in lift
coefficient prediction $E(c_l)$. Early investigations showed that doing so can yield a very high agreement between the expected
airfoil coefficient and the one based on interpolation from the sensor positions. However, when looking at the resulting sensor
positions themselves, it appeared that the optimization routine had merely found a sensor layout which resulted in a close fit
in lift prediction while the pressure distribution was not at all captured well. It should be noted that $c_{p,int}(\alpha,s)$ is derived
using linear interpolation/extrapolation. Using higher-order interpolation schemes could potentially increase the accuracy with
190 which the pressure distribution is approximated, but could also introduce numerical artifacts undesired in the proof-of-concept
provided by this study.

This study, analyzes the effect of sensor placement on the lift prediction, specifically, though the technique could alternatively
be applied to improve the measurement of the pitching moment or the pressure component of the drag force. Potential other
objectives, such as the accurate determination of the angle of attack or the separation point, would necessitate alternative
formulations of the objective function considered outside of this article's scope.

### 2.4.3 Sequential quadratic programming layout optimization (SQP)

Another optimization algorithm, sequential quadratic programming (SQP), was implemented to ensure the robustness of solution for the GA described in the previous section. Kelley et al. (2023) showed the benefits of an SQP optimized port layout
including lift coefficient error reduction compared to cosine spacing. The number of pressure ports was reduced from 48 to 30





to measure lift coefficient with less than 5% error across a broad range of angles of attack for a NACA $64_3 - 618$ airfoil by using the SQP optimized layout instead of cosine spacing.

The SQP optimization algorithm is suited for constrained and non-linear problems. Details of SQP are well documented in Biggs (1975); Boggs and Tolle (2000). Design variables and the objective function of the SQP optimization are identical to the GA optimization approach in Section 2.4.2. This ensured any differences in the port location solutions were limited to the two optimization algorithms described. The SQP algorithm was directly swapped within the minimisation function call implemented for the GA approach. The GA and SQP layout optimization were both implemented in Matlab's Global optimization Toolbox.

### 2.4.4 Limiting the optimization algorithm

For the generic optimization problem presented in this study, a design variable space of $0 \leq p \leq 2$ is chosen. In an experiment, however, many practical reasons might limit the spacing of the sensors, a couple of which are discussed below:

- *Fixed sensor position*: If it is desired to fix one sensor at a specific location on the airfoil surface, say at the leading edge of an airfoil, the upper and lower bound of a design variable can be altered such that $p_1 = 1$, while the other design variables are free to be optimized in $0 \leq p \leq 2$.

- *Sensor size*: A real sensor has a finite size, e.g. the diameter of the pressure tap, and therefore, a minimum distance between sensors has to be ensured, which allows for their installation.

- *"No-go" zones*: If certain areas of the tested airfoil are inaccessible, the placement of a sensor in such a "no-go" zone can be avoided. This could be relevant for, e.g. a region at the trailing edge too thin to allow for the internal guidance of pressure tubes, the existence of trailing edge adhesive or the presence of internal structures such as a shear web.

The above constraints can be readily applied in the SQP and GA optimization algorithms. While the first is related to input settings, the latter two can be enforced by outputting an unrealistically high value from the objective function if the desired criteria are not met. The optimization routine then does not converge towards layouts which violate the minimum sensor spacing or "no-go" zones.

## 3 Results

This section presents the results of applying cosine spacing and optimization routines to obtain the pressure sensor layout. For all approaches, numbers of sensors of $5 \leq N_s \leq 40$ are considered for the four FFA airfoils under investigation.

### 3.1 Cosine spacing

As mentioned in section 2.4.2, the optimization routines do not optimize for lift prediction accuracy but instead for an accurate representation of the pressure distribution. While this ensures that no cancellation of errors occurs, the accuracy of lift prediction is a direct consequence of a well-represented pressure distribution.



The quality of representation of the pressure distribution as a function of the number of sensors is shown in figure 7 (a) for cosine-spaced sensors. Irrespective of the investigated airfoil, this error initially falls sharply before entering a region in which the increase in the number of sensors barely affects the prediction quality. Figure 7 (b) depicts the resulting error in lift prediction. As with the error in the representation of the pressure distribution, an increase in sensors leads to a strong initial decrease of error before more gently decreasing for higher $N_s$. For $N_s \gtrsim 25$, the error of the predicted lift is $E_{prob}(c_l) \leq 0.01$.

For both the accuracy of pressure distribution and lift estimation, it becomes apparent that even numbers of sensors perform considerably better than odd numbers of sensors. This indicates that the steep pressure gradient at the leading edge can be captured accurately without a sensor placed exactly at the leading edge. Having two sensors close to (but not exactly at) the leading edge instead is beneficial for capturing the suction peak and stagnation point. This is the case for even numbers of sensors. This trend is lost upwards of $N_s \approx 30$ where the prediction error behaves more randomly.

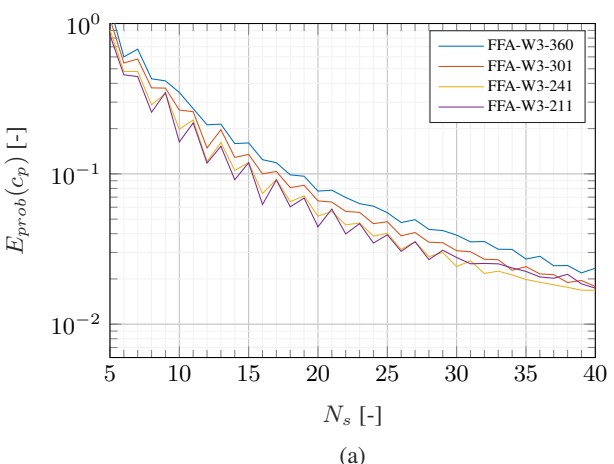
(a)

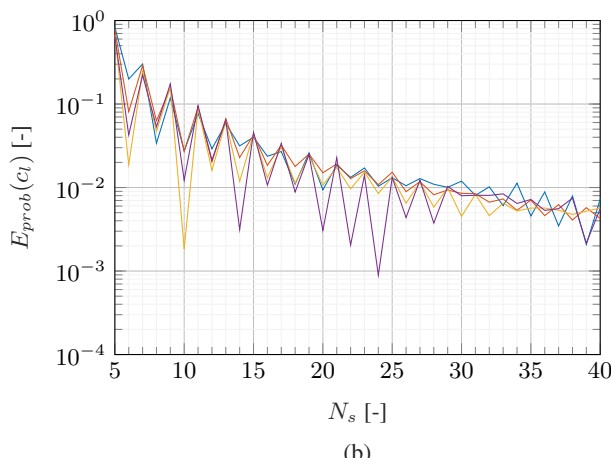
(b)

**Figure 7.** Error in the representation of the $c_p$-distribution (a) and $c_l$ determination (b) as a function of the number of sensors using a cosine sensor spacing

## 3.2   Optimized sensor layout

Based on their expected operating conditions, each investigated airfoil has a different range of expected angles of attack and, thus, an individual objective function. Additionally, the airfoil's pressure distributions differ significantly due to their range of relative thickness. Therefore, the optimization routines arrive at a sensor layout tailored to the individual airfoil. Figure 8 shows the optimized sensor layout for the four airfoils using $N_s = 15$ sensors. The individual plots contain the pressure distribution at
the angle of attack with the highest probability of occurrence, see also figure 5. Both optimization routines converge to almost identical sensor layouts. Furthermore, the optimized layouts capture individual features of the pressure distributions, such as the flow separation on the suction side of the FFA-W3-360 airfoil or the sharp suction and stagnation peaks of the FFA-W3-211 airfoil, very well.





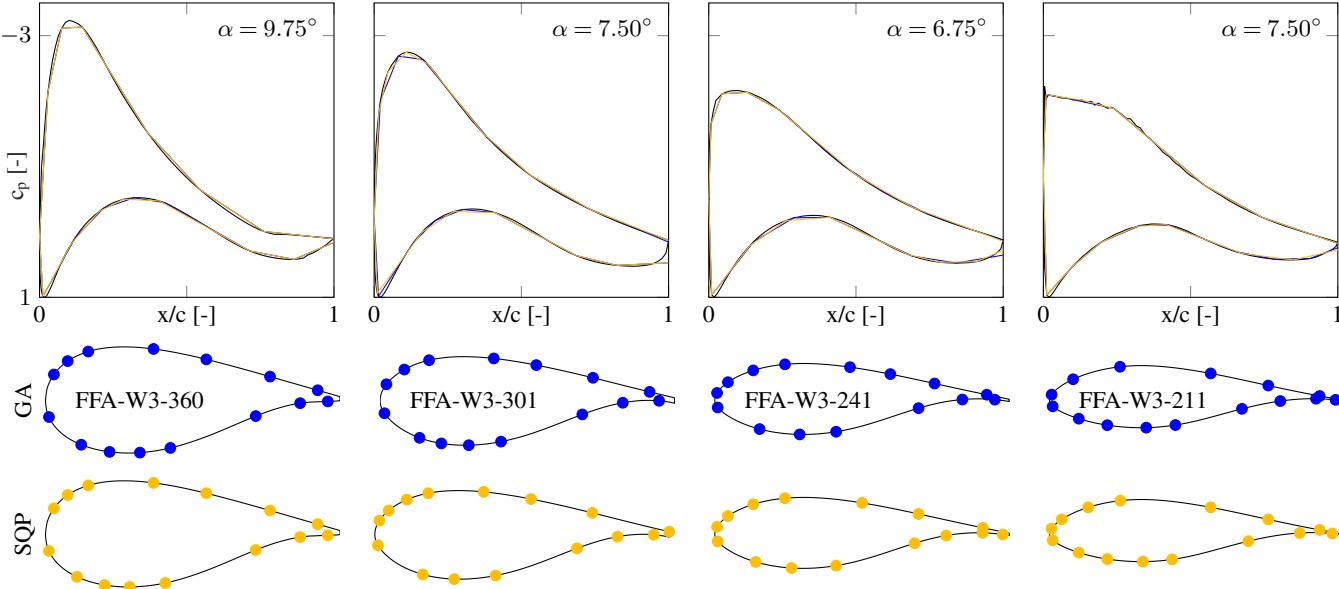

**Figure 8.** Optimized pressure sensor layouts for $N_s = 15$ along with the expected (black) and interpolated (blue and yellow) pressure distributions at the angle of attack with the highest probability of occurrence per airfoil

To further underline the advantage of sensor layout optimization, figure 9 shows both optimized layouts as well as the cosine-spaced counterpart for an increasing number of sensors on the FFA-W3-241. Again, the GA and SQP optimizers converge to almost identical results. It is evident that for lower $N_s$, the optimized layouts yield a much higher fidelity to the actual pressure distribution at the angle of attack with the highest probability of occurrence. While the optimized layouts achieve an almost perfect match for $N_s = 20$, there are still apparent deviations between the expected pressure distribution and that interpolated from a cosine spacing.

Given the similar convergence behaviour of the two optimization routines, only the results created using the genetic algorithm are considered from here on. The optimized layout's accuracy in predicting the pressure distribution and the lift coefficient as a function of the number of sensors is shown in figure 10. Comparing these results to the ones achieved using a cosine spacing, see figure 7, the optimized layout exhibits a higher accuracy for the same number of sensors.

The probability of specific angles of attack to occur drives the optimizer towards layouts allowing an accurate representation of the pressure distribution in the expected conditions. To further evaluate the benefit of layout optimization, the difference in errors between the optimized and cosine layout can be calculated for all individual angles of attack, thus also including those expected to occur less often. Figure 11 exemplarily shows this difference of errors for the FFA-W3-241 airfoil and varying numbers of sensors. The pressure distribution is clearly represented better when using an optimized layout. While there is an overall large improvement for very low numbers of sensors ($N_s = 5$), the largest reductions in error are found around the main




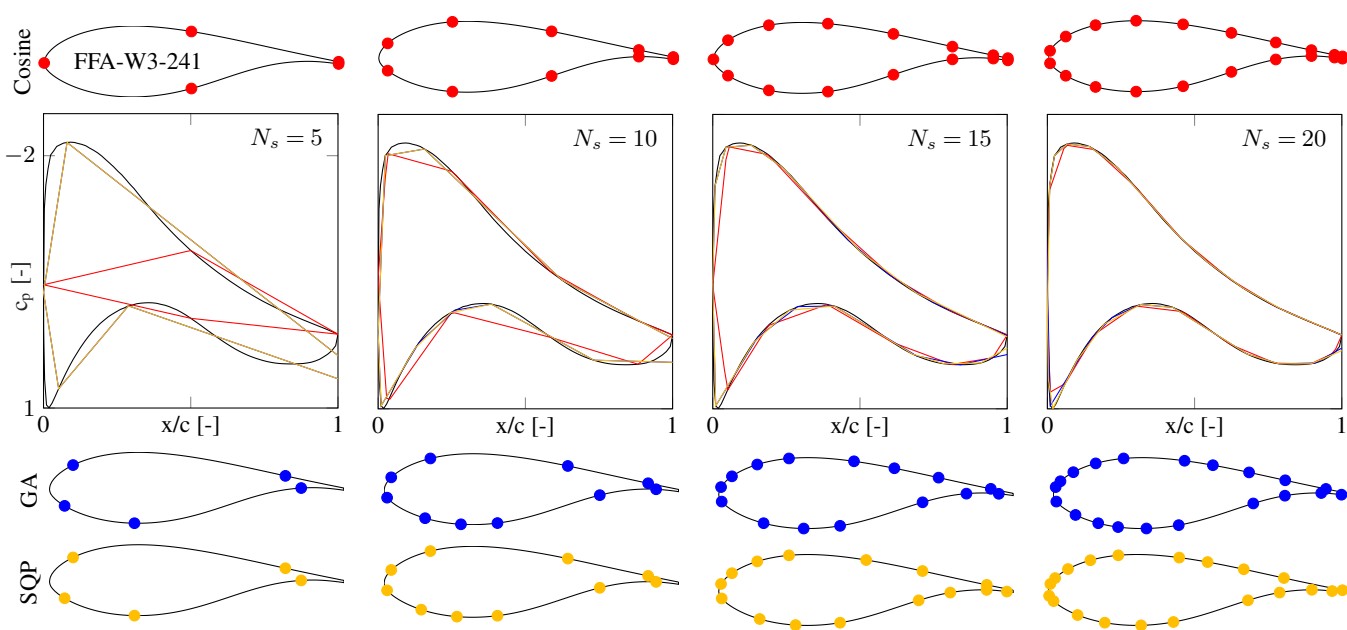

**Figure 9.** Accuracy in representing the expected (black) pressure distribution when using a cosine sensor spacing (red) and optimized layouts (blue and yellow) for a varying number of sensors, shown for the FFA-W3-241 airfoil and $\alpha = 6.75°$

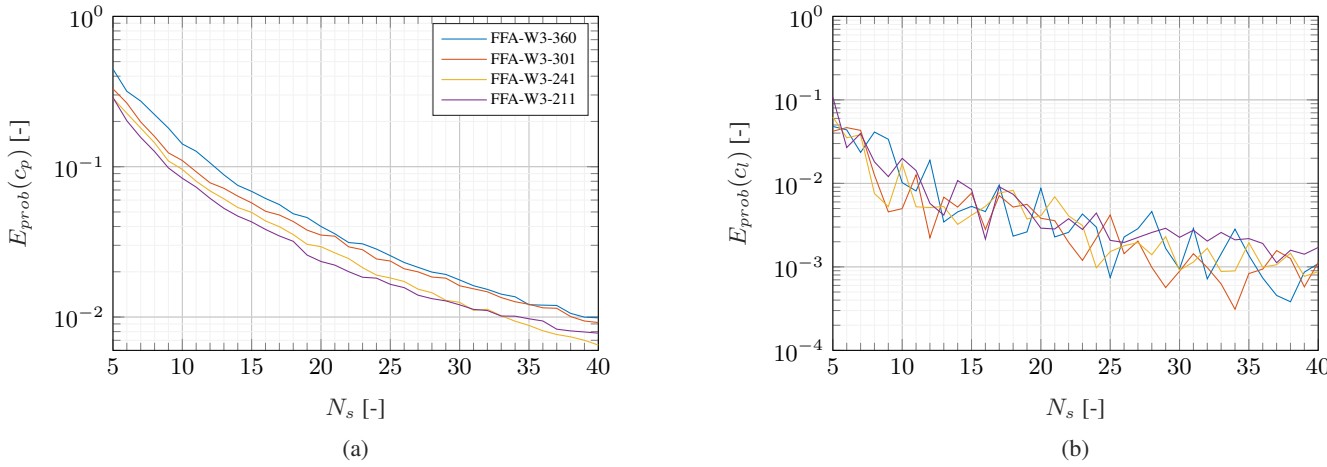

**Figure 10.** Error in the representation of the $c_p$-distribution (a) and $c_l$ determination (b) as a function of the number of sensors using a GA-optimized sensor layout

expected angle of attack ($\alpha = 6.75°$ for the FFA-W3-241 airfoil) for higher numbers of sensors. With increasing numbers of sensors, the error of optimized and cosine layout reduces and, consequently, their difference, too.



For positive angles of attack, the optimized layouts generally also outperform the cosine-spaced layout in predicting the lift coefficient. The exception is the cosine sensor layout with $N_s = 10$ sensors, which gives a very good approximation of the lift coefficient. Similar cases, where the cosine spacing yields very good lift predictions by means of error cancellation in the pressure distribution representation, also occur for the FFA-W3-211 airfoil for $N_s = 14, 20, 22, 24$. These cases are also visible in figure 7 (b) and should be interpreted as outliers.

This analysis of accuracy differences in lift prediction and pressure distribution representation shows that even though the optimization is driven by the angles of attack expected to occur most often, it has a positive impact throughout large ranges of angles.

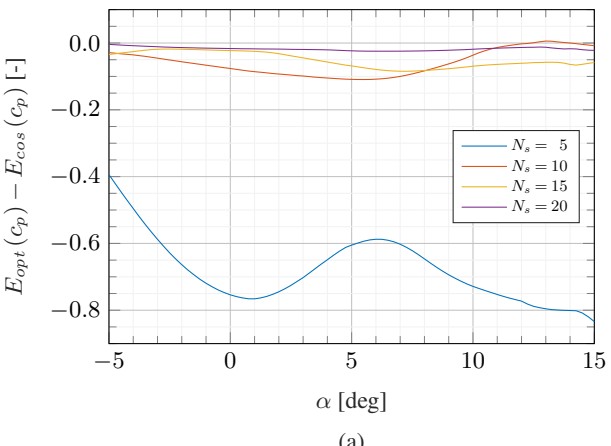
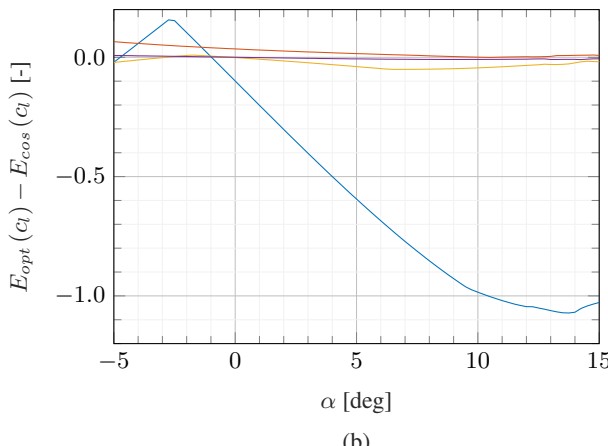

(a)                                          (b)

**Figure 11.** Difference of error in the representation of the $c_p$-distribution (a) and $c_l$ determination (b) between an optimized and cosine-spaced sensor layout as a function of angle of attack, shown for the FFA-W3-241 airfoil

## 3.3 Potential for reducing the number of sensors

To estimate the potential for reducing the number of sensors, power law curve fits are applied to all graphs shown in figures 7 and 10. This serves the purpose of capturing the general trends of how many sensors are required for a specific level of accuracy without the local maxima and minima present in the underlying curves. The parameters used in the individual curve fits following equation

$$N_s \left( E_{prob} \right) = A \, E_{prob}^{-B} \tag{16}$$

are listed in table 2.

Based on these curve fits, a ratio of optimized to cosine spaced sensors $N_{s,opt}/N_{s,cos}$ can be calculated as a function of a specified error in lift prediction or representation of the pressure distribution. Figure 12 shows this ratio of required sensors for targeted errors of $0.001 \leq E_{prob} \leq 1$.




| FFA-W3- | | Cosine spacing | | | | Optimized layout | | | |
|---|---|---|---|---|---|---|---|---|---|
| | | 360 | 301 | 241 | 211 | 360 | 301 | 241 | 211 |
| $E_{prob}(c_p)$ | A | 5.474 | 5.098 | 4.616 | 4.296 | 3.570 | 2.820 | 2.852 | 2.441 |
| | B | 0.522 | 0.512 | 0.516 | 0.540 | 0.527 | 0.575 | 0.536 | 0.570 |
| $E_{prob}(c_l)$ | A | 3.612 | 3.381 | 4.443 | 5.097 | 2.075 | 1.710 | 1.378 | 1.255 |
| | B | 0.428 | 0.441 | 0.350 | 0.312 | 0.399 | 0.415 | 0.463 | 0.509 |

**Table 2.** Parameters for curve fits

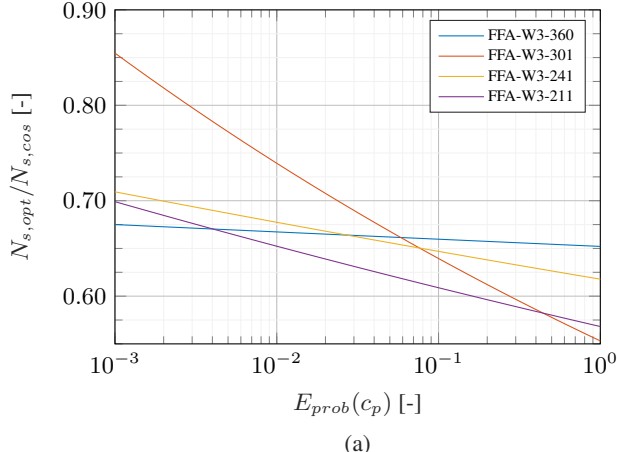
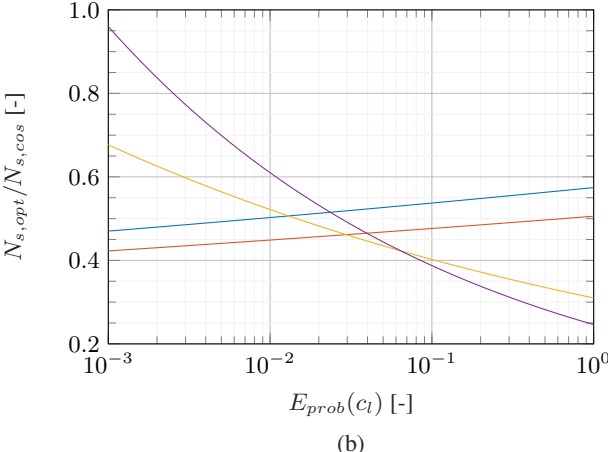

**Figure 12.** Ratio of required number of sensors between an optimized and cosine-spaced sensor layout to represent the pressure distribution (a) and the lift coefficient (b) with a specified accuracy

As expected, the number of sensors required to achieve a certain accuracy is always lower for the optimized layout than for the cosine-spaced layout. Exemplary, for a lift accuracy of $E_{prob}(c_l) = 0.01$, the ratio of required sensors lies between $N_{s,opt}/N_{s,cos} = 0.45$ and $N_{s,opt}/N_{s,cos} = 0.61$ depending on the airfoil, see figure 12 (b). Assuming that $N_s = 25$ sensors are required to achieve an accuracy of $E_{prob}(c_l) = 0.01$ with a cosine spacing, approximately ten to 14 sensors less yield the same accuracy when placed in an optimized layout.

Historically, experimental testing has been performed predominantly on thin airfoils and with many sensors. The analysis presented here demonstrates that the thinner airfoils are special beneficiaries of the optimization approach when fewer sensors are available but exhibit less of an advantage over the conventional cosine spacing for higher numbers of sensors. For thicker airfoils, sensor layout optimization has a more constant positive impact on lift prediction throughout the range of desired accuracies.



## 4    Conclusions

Pressure measurements are a commonly used measurement technique to aerodynamically characterize airfoils, in particular, to derive their aerodynamic loading. In most experiments, the accuracy of predicting aerodynamic properties is ensured by placing a large amount of pressure sensors on the investigated geometry. There are, however, situations which do not allow for the placement of such large numbers of sensors, e.g. due to geometrical, structural or financial restrictions. For these situations, the present work details a robust approach to optimize the pressure sensor layout for fidelity to the expected aerodynamic conditions. To this end, pre-calculated pressure distributions are input to two optimization routines, a genetic algorithm and a sequential quadratic programming algorithm, with the sensor locations as design variables. The pressure distributions are weighted based on the expected occurrence of angles of attack. The sensor layout optimization is applied to the generic case of the IEA 15 MW reference wind turbine, whose blades are defined by the FFA airfoil family. It is expected that the optimization approach is suited for other airfoil families as well.

The fact that two algorithms using fundamentally different optimization routines converge on almost identical sensor layouts suggests that an optimal solution exists for this problem. The optimized layouts show a clear advantage over a simpler layout using cosine spacing. They capture the expected pressure distribution more accurately and, consequently, allow a better approximation of the lift coefficient. Even though the optimization is driven by those angles of attack most likely to occur, the positive impact of sensor layout optimization is present for large ranges of angles of attack. Based on these benefits, fewer sensors are required in an optimized layout than in a cosine-spaced layout with the same accuracy. Depending on the targeted error in lift prediction as well as the regarded airfoil geometry, a 39-55 % reduction in the number of sensors compared to cosine spacing is achievable. As such, the presented optimization approach can contribute significantly to improving the data quality, reducing unnecessary equipment and saving cost in experimental setups. The port savings come mainly from the chordwise regions where the pressure coefficient is linear. This is usually located at the maximum thickness location on the suction surface of the airfoil, and the inflection point of airfoil shape on the pressure surface.

Cost-savings are particularly relevant in full-scale wind turbine blade aerodynamics measurements using pressure ports. Low numbers of pressure ports and transducers may be a low cost solution. The present work demonstrates the potential to use as few as 5-10 pressure ports and still achieve lift coefficient errors less than 10 % to 2 %, respectively, with an optimized port layout. Further reduction of lift coefficient error with very low numbers of pressure ports may be possible by adjusting the optimizer's objective function. Analysis in Kelley et al. (2023) minimized lift coefficient error as the objective function instead of the sum of pressure coefficient errors. The shape of the pressure coefficient curve was not well represented in the optimal solution because no ports were placed near the suction peak. However, the integration of pressure to lift coefficient was surprisingly accurate with less than 10 % lift coefficient error using only 8 ports for a large range of angles of attack. The potential of such minimalistic sensor layouts optimized for lift coefficient accuracy should be investigated in future research.

To further increase the robustness of the optimization approach presented here, future investigations should aim to incorporate aspects critical to experiments, such as sensor failure, measurement uncertainty, or a change of the airfoil's pressure distribution due to roughness development, into the optimization routine. Furthermore, the probability of specific angles of





attack to occur is calculated based on the assumption that a single angle of attack occurs per wind speed. In realistic condi-
330 tions, many characteristics, such as rotor tilt, yaw misalignment, wind shear, turbulence, etc., cause the angle of attack to vary
dynamically. These conditions could also lead to dynamic stall. These unsteady effects on optimal port placement are not part
of the existing work. But it would be interesting to observe whether the optimized sensor layouts change when adding more
realistic inflow and operating conditions to the methodology presented in this study.

*Code availability.* A script demonstrating the optimization routines presented in this study is openly available on the 4TU.ResearchData
repository at DOI:0.4121/99662eaf-ac79-4952-ad80-6d7de3708427.





## Appendix A: Nomenclature

**Latin letters**

| | |
|---|---|
| $A, B$ | Curve fitting parameters |
| $a, a'$ | Rotor averaged axial and tangential induction factor |
| $a_B, a'_B$ | Local axial and tangential induction factor at blade |
| $CDF$ | Cumulative distribution function |
| $C_{PDF}$ | Scaling factor |
| $C_T$ | Thrust coefficient |
| $c$ | Chord |
| $c_l, c_d, c_m$ | Lift, drag and moment coefficient |
| $c_n, c_t$ | Chord normal and tangential force coefficient |
| $c_p$ | Pressure coefficient |
| $E$ | Error function |
| $F_{tip}, F_{root}$ | Prandtl root and tip correction factors |
| $GA$ | Genetic algorithm |
| $N_b$ | Number of blades |
| $N_s$ | Number of pressure sensors |
| $N_\alpha$ | Number of investigated angles of attack |
| $\boldsymbol{n}$ | Normal vector |
| $P$ | Probability |
| $PDF$ | Probability density function |
| $p$ | Optimization design variable (chordwise sensor position) |
| $R$ | Blade tip radius |
| $\mathrm{Re}_c$ | Chord Reynolds number |
| $r$ | Radial coordinate |
| $r_{root}$ | Blade root radius |

| | |
|---|---|
| $s$ | Airfoil surface coordinate |
| $SQP$ | Sequential quadratic programming |
| $t$ | Airfoil thickness |
| $U_{ave}$ | Average free stream velocity according to IEC standard 61400-1 |
| $U_{ref}$ | Reference wind speed average over 10 min according to IEC standard 61400-1 |
| $U_\infty$ | Free stream velocity |
| $V_{eff}$ | Local inflow velocity |
| $x$ | Chordwise coordinate |

**Greek letters**

| | |
|---|---|
| $\alpha$ | Angle of attack |
| $\beta_{pitch}$ | Blade pitch angle |
| $\beta_{twist}$ | Blade twist angle |
| $\lambda$ | Tip speed ratio |
| $\lambda_r$ | Local tip speed ratio |
| $\mu$ | Dynamic viscosity of air |
| $\rho$ | Density of air |
| $\phi$ | Inflow angle |
| $\omega$ | Angular velocity |

**Subscripts**

| | |
|---|---|
| $cos$ | Cosine sensor layout |
| $exp$ | Expected true value |
| $int$ | Interpolated |
| $opt$ | Optimized sensor layout |
| $prob$ | Weighted by each angle of attack's probability of occurrence |

*Author contributions.* EKF: Conceptualization, methodology, investigation, writing; CLK: Conceptualization, methodology, reviewing, editing; KAB: Conceptualization, methodology, reviewing, editing





*Acknowledgements.* This contribution has been financed with Topsector Energiesubsidie from the Dutch Ministry of Economic Affairs under grant no. TEHE119018.

Sandia National Laboratories is a multimission laboratory managed and operated by National Technology & Engineering Solutions of Sandia, LLC, a wholly owned subsidiary of Honeywell International Inc., for the U.S. Department of Energy's National Nuclear Security Administration under contract DE-NA0003525.



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
