# Peer review of "On optimizing the sensor spacing for pressure measurements on wind turbine airfoils"

_Wind Energy Science, 2024_

## Author Comment (AC1)

Dear Prof. Bayoán Cal,

Thank you for giving us the opportunity to submit a revised version of the manuscript titled "On optimizing the sensor spacing for pressure measurements on wind turbine airfoils" to *Wind Energy Science*. We appreciate the time and effort that you and
5  the reviewers have dedicated to providing valuable feedback on our manuscript. We have been able to incorporate changes to reflect the suggestions provided by the reviewers. Please find a point-by-point response to their comments below.

**Reviewer 1**

*Unnumbered comment*:
For the future work it would be interesting to write full CFD analysis version of the paper. Xfoil is a great tool but it has its
10  limitations.
We fully agree with your comment. Employing CFD could be a valid alternative to obtain the pressure distributions used as input for the optimization routines. It would be interesting to compare the resulting sensor layouts. The choice for XFOIL was made to facilitate the replication of the results we present here as the tool is widely used, and its inputs are subject to fewer intricacies than those of CFD simulations would likely be.

15  1. Refer to line 55: Please explain the rationale behind the specific choice of these two optimization schemes among others available should be explained. What are the competitive advantages of using these two optimization schemes?
Genetic algorithms do not require derivative information and have a good chance of converging towards the global optimum due to searching the entire design space. As such, they are well-suited for a relatively complex optimization problem as posed in this paper. The sequential quadratic programming algorithm, on the other hand, is gradient-based
20  and, thus, more prone to get stuck in a local minimum. Comparing the results of both methods can be indicative of whether multiple minima exist or whether there is one clear optimum. An advantage of the SQP is its computational efficiency. We have added statements along this line of argumentation in the sections introducing the respective optimization routines, see Page 9, Line 184 and Page 10, Line 215.

2. Refer to line 74: Explain what the authors mean by blended and non-blended profiles. Does blending refer to combin-
25  ing two or more standard airfoils to achieve a more aerodynamically optimized airfoil? This question arises because the authors have specifically chosen non-blended profiles, even though the optimization procedure they present seems applicable to blended profiles as well. This specific choice suggests that there may be particular characteristics unique to non-blended profiles.
The blended airfoils are part of the IEA 15 MW RWT's documentation. While this is not explicitly mentioned in the IEA
30  15 MW RWT report, they seem to be airfoils created by interpolating the airfoil geometries of standard FFA-W3 airfoils based on relative thickness for the purposes of creating a smoothly lofted blade surface. For example, the FFA-W3-270blend airfoil is then the geometry resulting from interpolating the geometries of the FFA-W3-301 (30.1 % thickness) and FFA-W3-241 (24.1 % thickness) airfoils for a thickness of 27 %. Given that they are not part of the original FFA-W3 airfoil family, they were excluded from the analysis. However, as you comment yourself, the analysis presented in this
35  study could have been applied to the blended airfoils just as well. In conclusion, we base our airfoil selection on the fact that these airfoils are part of the original FFA-W3 airfoil family and are well-documented. We have added a statement along those lines, see Page 3, Line 74.

3. Refer lines 97 and 98: Please reference any papers that demonstrate tangential induction has minimal impact within the chosen range of r/R for this study. Including this evidence would enhance the article, as this assumption is crucial in
40  determining the angle of attack.
We want to acknowledge that you are raising a valid point by questioning the zero tangential induction assumption. As far as we are aware, no papers have reported the tangential induction distribution of the IEA 15 MW RWT so far. We are aware that a multi-fidelity numerical benchmark of this turbine is currently underway in IEA Wind Task 47, but the results have not yet been published. To make the reader aware that this assumption should be considered

45 carefully, particularly towards the root, we have adjusted the text, see Page 4, Line 98. We would like to refrain from including numerical simulations (beyond the 1D momentum computations) ourselves here, as their description would add considerable complexity to the methodology and decrease the replicability of the presented research. We hope you can agree that this approach is accurate enough for this proof of concept study. We carried out blade element momentum theory calculations where exact tangential induction was known for a blade with similar performance to the IEA 15 MW

50 RWT, this assumption leads to $0.1°$ error in angle of attack at the blade mid-span, and $0.01°$ error in angle of attack at the blade tip. Therefore the zero tangential induction approximation is sufficient for 2D airfoil analysis in the pressure port layout optimization that is representative of the airfoils on the IEA 15 MW RWT in operation.

4. Suggestion for the equation 10: The Euclidean dot product between a scalar and a vector is equivalent to multiplying the scalar throughout the vector. Therefore, $c_p \cdot n(s)$ can be simplified to $c_p n(s)$. Thus, the dot product is redundant in this

55  context.

Thank you for this suggestion. We have adjusted the equation according to your suggestion, see Equation 10.

5. Refer to the equation 14: I suggest that the absolute value of $(C_{l,int}(\alpha) - C_{l,exp}(\alpha))$ should be included in the equation. Without the absolute values, positive and negative error values may cancel each other out when summing across the range of angles of attack, which could obscure the true magnitude of the error. If the omission of the modulus sign is

60  merely a typographical error, that is acceptable; however, if the equation has been applied as it appears in the paper, I am concerned it may not accurately reflect the error values and trends depicted in Figures 7(b) and 10(b). Please calculate $E_{prob}(c_l)$ using the absolute value $|(C_{l,int}(\alpha) - C_{l,exp}(\alpha))|$ and include this comparison in your response to this review.

Thank you for pointing this out. It is indeed a typographic error and in the optimization routines, absolute errors were used to avoid cancellation of errors. We have adjusted Equation 14 and also Equation 13, where the same error was

65  made.

6. Suggestion for the equation 15: It would be more appropriate if the authors also placed 'min' on the right-hand side of the equation, given that they are minimizing this function. Alternatively, if the authors prefer to include 'min' on only one side, they should enclose the entire equation in brackets after 'min'.

You are right, this is more appropriate. We have added a 'min' on the right side of the equation, see Equation 15.

70 **Reviewer 2**

1. As the estimated pressure Cd is very sensitive on the used (discrete) pressure distribution, a note on the predicted Cd from the optimised distribution would be very informative and enhance the understanding of the usefulness of the optimised distribution. It is mentioned as future work, but just a small paragraph/figure to show how bad/good it is would be very beneficial.

75 Thank you for this suggestion. We have added a short discussion of the influence of sensor layout optimization on the accuracy of drag determination, see Page 14, Line 290ff. The added analysis suggests that sensor layout optimization also improves the drag prediction for most cases.

2. Eq. 2: Check these expressions, I think there is a typo; the factor on the exp-function should be $\pi * U_{inf}/U_{avg}^2$?

We have re-checked both the referenced IEC standard, which explicitly states the CDF as given in Eq. (1) as well as the

80 derivation leading to Eq. (2). The checks lead to the same equations as currently presented, so we have left them as they are.

3. Line between Eq (3) and (4), give the value for CT2 as you do for Ct1

We have added the value, see Page 4, Line 92.

4. Figure 3: Do you include the controller in the simulations?

85 No controller is used as we do not run full BEM simulations. Instead, we make use of 1D momentum theory to solve for the axial induction terms. Next to the technical report, the IEA 15 MW RWT documentation comes with an excel sheet

that lists the rotor performance as a function of wind speed. This includes values of the rotor speed, thrust coefficient and pitch angle, which allows the calculation of the angle of attack as listed in Equations (3) – (7). We believe that our approach is sufficiently described at Page 4, Line 87ff. We hope you can agree with this assessment.

5. Table 1: High Re, is this an issue for XFoil?
   Previous research used the experimental results obtained in the AVATAR project to validate XFOIL for high Reynolds numbers. This experiment characterized a DU00-W-210 airfoil in a pressurized wind tunnel at a Reynolds number 15e6, thus, comparable flow conditions and airfoil thickness to the outboard airfoils studied here. In this validation exercise, XFOIL performed reasonably well, giving us confidence in its use in the present study. We have included a statement along those lines as well as relevant references, see Page 6, Line 131.

6. Section 2.4.2: I am missing a reference to a general description of the GA algorithm
   We have added a reference to the description of the GA, see Page 9, Line 186.

7. Eq (15): Perhaps emphasise that the objective function is an integral and not a summation of the pressure difference at the discrete points, where the objective function would be zero. I was a little confused during the first read through, as my mind was focused on a discrete distribution. It is a good comment about the potential of improving the Cl prediction by using other interpolation functions.
   We see your point of how this could be confusing to the reader. Thank you for pointing that out to us. We have added a statement to clarify this for the reader, see Page 9, Line 194.

We would like to thank the reviewers for their detailed and constructive feedback. Their comments have been very helpful in improving the quality of our manuscript. Please find attached a version of our manuscript highlighting all the changes made. We look forward to hearing from you in due time regarding our submission and to responding to any further questions and comments you may have.

Sincerely,

Erik Fritz, Christopher Kelley, Kenneth Brown

[revised manuscript text omitted]